# Magnetic Resonance Imaging of Submental and Masticatory Muscle Morphology and Its Relationship with Temporomandibular Joint Structures

**DOI:** 10.3390/diagnostics15121535

**Published:** 2025-06-17

**Authors:** Melisa Öçbe, Mahmut Sabri Medişoğlu

**Affiliations:** 1Department of Oral and Maxillofacial Radiology, Faculty of Dentistry, Kocaeli Health and Technology University, Yeniköy Mahallesi Ilıca Caddesi No:29, Başiskele 41275, Kocaeli, Turkey; 2Department of Anatomy, Faculty of Dentistry, Kocaeli Health and Technology University, Yeniköy Mahallesi Ilıca Caddesi No:29, Başiskele 41275, Kocaeli, Turkey; sabri.medisoglu@kocaelisaglik.edu.tr

**Keywords:** masticatory muscles, submental muscles, temporomandibular joint, muscle thickness, MRI, TMJ disorders, disc displacement

## Abstract

**Introduction**: This study aimed to evaluate the submental and masticatory muscles in patients of different age groups using magnetic resonance imaging (MRI) and computed tomography (CT) methods, and investigate potential associations between muscle morphology, temporomandibular joint (TMJ) structures, and disc displacement. **Materials and Methods**: A total of 185 MRI scans were retrospectively analyzed to assess the thickness of the digastric, geniohyoid, mylohyoid, medial pterygoid, masseter, and lateral pterygoid muscles bilaterally. TMJ hard tissue changes were classified using computed tomography (CT). Correlations between muscle thickness and TMJ structures were analyzed using Pearson correlation coefficients, with statistical significance set at *p* < 0.05. **Results**: The study population included 110 females and 75 males, with a mean age of 50.08 ± 20.15 years. The largest age group was 51–75 years (41%), followed by 18–35 years (28%). Significant correlations were observed between muscle thickness and TMJ structures as follows: Right digastric muscle showed a significant association with right disc–condyle position (*p* = 0.02). Right mylohyoid muscle exhibited a strong correlation with right disc–condyle position (*p* = 0.004). Left medial pterygoid muscle was significantly correlated with left condyle pathology (*p* = 0.02). Left masseter muscle showed a significant correlation with left condyle pathology (*p* = 0.014). Condylar flattening was the most frequent pathology, observed in 58% of right condyles and 53% of left condyles. Disc displacement was present in 41% of right TMJs and 34% of left TMJs. **Conclusions**: This study highlights the important associations between masticatory and submental muscle morphology and TMJ structures, suggesting that muscle function may play a role in condylar positioning and disc alignment. These findings emphasize the need for comprehensive muscle evaluation in TMJ disorder (TMD) diagnosis and treatment planning.

## 1. Introduction

The masticatory and submental muscles play a critical role in both jaw movement and swallowing function. The submental muscles, including the mylohyoid, anterior belly of the digastric, and geniohyoid muscles, are responsible for hyoid elevation and jaw opening [1]. Their coordinated contractions contribute to swallowing by facilitating the anterior–superior movement of the hyolaryngeal complex and the opening of the upper esophageal sphincter [2,3]. Dysfunctions in these muscles can be associated with swallowing disorders, temporomandibular joint (TMJ) disorders [4,5], and compromised oral function, particularly in aging populations affected by frailty. In addition, aging is associated with progressive muscle loss, known as sarcopenia, which affects both skeletal and craniofacial muscles, including the submental and masticatory muscles. Previous studies have demonstrated that sarcopenia-related changes in these muscles may impact swallowing function, jaw mobility, and temporomandibular joint dynamics [5,6,7,8]. Given these potential influences, this study categorized participants into age groups to evaluate whether muscle thickness varies with aging and to assess its potential relationship with TMJ structures [1,3,4].

Temporomandibular disorders (TMDs) are characterized by pain, joint sounds, mandibular movement limitations, and muscle sensitivity [4,5]. While previous studies have extensively investigated the role of masticatory muscles [4,5,6,7] in TMDs, the impact of submental muscles remains largely unexplored. These muscles act in close coordination with the cervical and masticatory muscles, forming a functional unit crucial for maintaining physiological oral function [2,3]. Dysfunction in one of these components can provoke compensatory changes in others, further complicating the clinical presentation of TMDs.

Despite their importance, imaging of the submental muscles is not commonly performed in oral and maxillofacial radiology. Previous studies have primarily used ultrasonography, sonographic elastography or electromyographic to evaluate these muscles [8,9,10], while magnetic resonance imaging (MRI) studies remain rare. Given the significance of submental muscles not only in relation to TMDs but also in preoperative planning for floor-of-the-mouth surgeries [11,12,13], advanced imaging techniques are essential for comprehensive assessment. This study aims to evaluate the submental and masticatory muscles in patients across different age groups using MRI, while also investigating potential relationships between TMDs, disc displacement, and TMJ hard tissue changes observed in computed tomography (CT) images.

## 2. Materials and Methods

This retrospective study was approved by the Institutional Review Board (IRB) (XXX University, Non-invasive Clinical Research Ethics Committee (Project No: XXX)) and conducted in accordance with local ethical regulations. Informed consent was obtained from all participants before inclusion. All patient data were anonymized to maintain confidentiality. The procedures used in this study adhere to the tenets of the Declaration of Helsinki.

### 2.1. Study Population

A total of 185 MRI scans of patients were evaluated. CT images were included for assessing TMJ hard tissue changes. Participants were categorized into different age groups to analyze potential age-related changes in muscle morphology.

Inclusion criteria for this study was as follows: Patients aged 18 years and older, patients who underwent MRI for TMD evaluation, patients with available CT scans for TMJ hard tissue assessment, no history of systemic neuromuscular diseases affecting muscle function. Exclusion criteria was as follows: History of previous TMJ or oral surgery affecting muscle morphology, neuromuscular disorders or systemic conditions impacting muscle integrity, inadequate MRI or CT image quality preventing accurate measurements.

### 2.2. Imaging Protocol and Analysis

MRI and CT scans were analyzed on a GE Healthcare Workstation, enabling the viewing of CT and MRI images in overlay mode across three planes simultaneously. The Revolution™ EVO/Optima™ CT660 CT Scanner (GE Healthcare Japan Corporation, Tokyo, Japan) was used for CT scans, and the SIGNA™ Pioneer (GE Medical Systems, Waukesha, WI, USA) was used for MR imaging.

MRI scans were acquired using fast spin echo (FSE) sequences. T1-weighted (T1W) coronal images were used to assess the thickness of submental muscles. Muscle thickness was measured in millimeters (mm), with reference points standardized for consistency (Figure 1). The anatomical landmarks selected for the measurement included:

#### 2.2.1. Anterior Belly of Digastric

The anterior belly of the digastric muscle thickness was measured bilaterally, below the lateral inferior border of the mandible. The selected measurement site was at the thickest portion of the anterior belly, positioned symmetrically on both sides of the midline. The location was verified to exclude overlapping structures such as the mylohyoid and subcutaneous tissues.

#### 2.2.2. Geniohyoid

The geniohyoid muscle thickness was measured at the midline, directly inferior to the mandibular symphysis, at the level where the muscle extends from the inner surface of the mandible to the hyoid bone. The measurement point was selected at the widest portion of the muscle belly. This area corresponds to the anterior submental region.

#### 2.2.3. Mylohyoid

The mylohyoid muscle thickness was assessed in the region forming the floor of the mouth, inferolateral to the geniohyoid muscle. The measurement was taken at the central portion of the muscle, ensuring it was perpendicular to the muscle fibers. This location was identified as the area with the most uniform cross-section, and the selected measurement points were equidistant from the midline.

T1W and T2W axial images were used to evaluate the masticatory muscles bilaterally (Figure 2). Measurements were taken at predefined anatomical landmarks to ensure reproducibility, as follows:

#### 2.2.4. Masseter Muscle

The masseter muscle thickness was measured at its widest portion, perpendicular to the muscle fibers. The measurement site was selected at the level of the mandibular ramus, ensuring it corresponded to the thickest part of the muscle belly. Bilateral measurements were taken at approximately the same horizontal plane to ensure symmetrical evaluation.

#### 2.2.5. Medial Pterygoid Muscle

The medial pterygoid muscle thickness was measured at its broadest region, located medially to the mandibular ramus, adjacent to the masseter muscle. The measurement was positioned to avoid overlap with adjacent soft tissues or the lateral pterygoid muscle.

#### 2.2.6. Lateral Pterygoid Muscle

The lateral pterygoid muscle thickness was measured at its widest portion, perpendicular to the muscle fibers. The measurement site was selected at the level of the infratemporal fossa, ensuring consistency across subjects. The muscle was identified anterior to the medial pterygoid, extending horizontally from the lateral aspect of the lateral pterygoid plate to the condylar neck of the mandible. Measurements were taken bilaterally, ensuring they corresponded to the superior and inferior heads of the lateral pterygoid muscle, avoiding overlap with the adjacent temporalis or medial pterygoid muscles.

All measurements were conducted using standardized digital calipers within the MRI viewer software (v1.0.34.1). The thickness was recorded in millimeters, and each measurement location was decided and confirmed by one oral and maxillofacial radiology specialist with 7 years of experience and one anatomy specialist with 23 years of experience.

Sagittal T1W and T2W images were analyzed to determine the relationship between the TMJ condyle and articular disc (Figure 3).

The presence of anterior disc displacement (ADD) was classified using the Research Diagnostic Criteria for Temporomandibular Disorders (RDC/TMD) [5], considering the position of the disc relative to the condylar head and articular eminence in closed-mouth positions. Disc displacement is based on the position of the articular disc relative to the condylar head and articular eminence in the closed-mouth position. Anterior disc displacement is diagnosed when the disc is positioned anterior to the condyle. CT images were examined for TMJ hard tissue changes and to classify condylar morphology based on established radiographic criteria (Figure 4):

#### 2.2.7. No Pathological Changes

The condylar head demonstrated a smooth, well-defined cortical outline. No signs of degenerative changes, bone irregularities, or remodeling were observed.

#### 2.2.8. Flattening

Loss of the normal convex shape of the condylar head, resulting in a broader and flattened condylar surface.

#### 2.2.9. Erosion

Presence of irregular, ill-defined cortical bone loss on the condylar head.

#### 2.2.10. Osteophyte Formation

Presence of bony outgrowths (osteophytes) along the condylar margins, indicative of advanced degenerative joint disease.

All evaluations were conducted by two independent observers, and discrepancies were resolved through consensus.

### 2.3. Statistical Analysis

Descriptive statistics, including mean, minimum, maximum, and standard deviation, were calculated for muscle thickness measurements across gender and age groups. Age was categorized into four groups: 18–35, 36–50, 51–75, and 76–100 years, and gender-based comparisons were performed to assess potential differences in muscle morphology. Pearson correlation analysis was conducted to evaluate relationships between muscle thickness and TMJ structures, including disc–condyle position and condylar morphology. Correlation coefficients (r-values) were calculated to determine the strength and direction of associations, while *p*-values were computed using the Student’s t-distribution to assess statistical significance. To examine muscle symmetry, right–left differences were calculated for all bilateral muscle measurements, and the distributions of these differences were analyzed. Additionally, the frequency of condylar morphological changes (flattening, erosion, and osteophyte formation) and disc displacement was assessed and compared between right and left TMJs. All statistical analyses were performed using Python v3.9 (Pandas, SciPy). A significance threshold of *p* < 0.05 was applied for all tests. Scatter plots were generated to visually explore relationships between muscle thickness, age, and TMJ parameters, and correlation matrices were constructed to identify key statistical relationships. To assess intra-observer reliability, muscle thickness measurements were taken at two different time points: an initial measurement and a repeated measurement after three months. Pearson correlation analysis was performed to determine the consistency between the two measurements for each muscle. The results demonstrated a high level of agreement, with correlation coefficients ranging from 0.9959 to 1.0000, all of which were statistically significant (*p* < 0.05).

## 3. Results

This study aimed to assess the thickness of submental and masticatory muscles in relation to TMJ structures using MRI and CT. The thickness measurements of the digastric, geniohyoid, mylohyoid, medial pterygoid, masseter, and lateral pterygoid muscles were obtained bilaterally in both male and female participants. These measurements were analyzed to determine potential differences based on gender and condylar morphology, as well as their relationship with TMJ disc position. MRI-based evaluations were performed using T1W and T2W coronal, sagittal and axial sections, allowing for precise muscle thickness assessment. Additionally, CT images were reviewed to classify condylar morphological changes, including flattening, erosion, and osteophyte formation. The relationship between muscle thickness and TMJ structures was statistically analyzed to identify significant correlations (Figure 4 and Figure 5).

A total of 185 participants were included in the study, with age distribution ranging from 18 to 92 years (110 female and 75 male). The study population was categorized into four age groups: 18–35 years, 36–50 years, 51–75 years, and 76–100 years. The largest group was 51–75 years (74 individuals, 41%), followed by 18–35 years (50 individuals, 28%). The 36–50 age group included 33 individuals (18%), while the smallest group was 76–100 years (19 individuals, 10%). Mean age was found as 50.08 (standard deviation: 20.15). The analysis of muscle measurements across different age groups reveals distinct trends. The 18–35 age group exhibited the lowest mean values for most muscle measurements, with right digastric (6.63), left digastric (6.55), and geniohyoid (9.81) showing relatively lower values compared to older age groups. In the 36–50 group, there was an increase in muscle thickness, particularly in geniohyoid (11.70), medial pterygoid (right: 15.10, left: 15.97), and lateral pterygoid (right: 15.1, left: 15.9), suggesting a peak in muscle mass during this period. The 51–75 group had slightly reduced values, with right digastric (7.59), left digastric (7.97), and geniohyoid (11.67) maintaining relatively stable levels. However, a decline was observed in medial pterygoid (right: 13.79, left: 13.75) and lateral pterygoid (right: 13.81, left: 13.77), indicating muscle loss with aging. The 76–100 group showed the most pronounced decline in muscle thickness, particularly in masseter (right: 12.94, left: 13.79) and lateral pterygoid (right: 12.64, left: 13.26), suggesting age-related atrophy. These findings indicate that muscle measurements peak around midlife and decline progressively with aging, with the most significant reductions observed in the oldest age group. The analysis of variance (ANOVA) was conducted to determine whether there were statistically significant differences in muscle thickness across the age groups. The results indicated that left digastric (F = 3.51, *p* = 0.017), geniohyoid (F = 4.66, *p* = 0.004), and right mylohyoid (F = 3.29, *p* = 0.022) showed statistically significant differences between age groups (*p* < 0.05). This suggests that muscle thickness in these regions varies significantly with age. In contrast, right digastric (F = 1.99, *p* = 0.118) and left mylohyoid (F = 0.62, *p* = 0.605) did not show significant differences, indicating that their thickness remains relatively stable across age groups. The means for each muscle thickness measurement are presented in Table 1, while the mean, minimum, and maximum values categorized by age groups and categorized by gender are presented in Table 2.

The statistical analysis of the correlation matrix revealed significant associations between certain muscle measurements and TMJ parameters. Specifically, a statistically significant correlation (*p* < 0.05) was observed between the right digastric muscle and the right disc–condyle position (*p* = 0.02), suggesting a relationship between the muscle’s function and disc alignment on the right side. Additionally, the right mylohyoid muscle demonstrated a significant correlation with the right disc–condyle position (*p* = 0.004). Variations in mylohyoid thickness might be associated with disc positioning.

On the left side, a significant correlation was noted between the left medial pterygoid muscle and the left condyle position (*p* = 0.02), suggesting that medial pterygoid muscle characteristics may influence condylar positioning. Furthermore, the left masseter muscle exhibited a significant correlation with the left condyle position (*p* = 0.014), implying a relationship between masseter function and condylar morphology (Table 3).

In the right condyle, 52 patients (28%) exhibited no pathological changes, while in the left condyle, 53 patients (28%) showed no abnormalities. Flattening of the condyle was the most common finding, observed in 108 patients (58%) on the right side and 99 patients (53%) on the left side. Erosion was present in 25 patients (13%) on the right and 32 patients (17%) on the left. Osteophyte formation was not observed in any patient on the right side, whereas it was detected in one patient on the left side. Regarding disc displacement, 77 patients (41%) exhibited displacement in the right TMJ, while 63 patients (34%) had displacement in the left TMJ.

The scatter plots in Figure 5 compare muscle thickness measurements between the right and left sides for the digastric, mylohyoid, medial pterygoid, and masseter muscles. The data points demonstrate a strong positive correlation, indicating bilateral symmetry in muscle thickness across subjects. This suggests that, in most cases, the thickness of these muscles is similar on both sides, with minimal asymmetry.

In Figure 6, the scatter plots illustrate the relationship between age and muscle thickness for various muscles, including the digastric, geniohyoid, mylohyoid, medial pterygoid, and masseter. The distribution of data points reveals variability in muscle thickness across different age groups, with some muscles showing potential trends related to aging.

## 4. Discussion

### 4.1. Role of Submental and Masticatory Muscles in Oral Function and Movement

The submental and masticatory muscles are essential for jaw movement, swallowing, and oral function. The submental muscle group, consisting of the mylohyoid, geniohyoid, and digastric muscles, primarily facilitates hyoid elevation, jaw opening, and oral floor stabilization. Among them, the mylohyoid muscle serves as a key structural component of the oral floor, influencing mastication, swallowing efficiency, and infection spread in conditions such as Ludwig’s angina [2,3,12,14]. The geniohyoid muscle aids in jaw depression and hyoid elevation, playing a critical role in swallowing initiation [14,15]. The digastric muscle, with its anterior and posterior bellies, coordinates jaw opening and hyoid movement, integrating with the other submental muscles for speech and deglutition.

The masticatory muscles, including the masseter, medial pterygoid, and lateral pterygoid, are primarily responsible for jaw stability and movement. The masseter muscle, originating from the zygomatic arch, functions as the primary mandibular elevator, generating significant bite force. The medial pterygoid muscle assists the masseter in jaw elevation and lateral movement, contributing to chewing efficiency. The lateral pterygoid muscle, with its superior and inferior heads, facilitates jaw protrusion, depression, and lateral excursion, playing a unique role in complex jaw mechanics and TMJ function. Dysfunction in these muscles has been linked to TMJ disorders, occlusal imbalances, and joint instability [14,15].

### 4.2. Age-Related Changes in Submental and Masticatory Muscles

Aging induces sarcopenia, leading to progressive declines in muscle mass, strength, and function. This deterioration is particularly evident in submental muscles, where muscle atrophy contributes to dysphagia, reduced jaw-opening strength, and inefficient swallowing [14,15]. Barotsis et al. identified geniohyoid muscle thickness below 0.65 mm as a predictor of sarcopenia, highlighting its potential as a diagnostic marker for swallowing dysfunction [16]. Similarly, Miura et al. found a positive correlation between geniohyoid thickness and jaw-opening strength, reinforcing the role of submental muscles in maintaining oral function [17].

The masticatory muscles also undergo age-related changes, with reductions in masseter and medial pterygoid thickness leading to weaker bite force, compromised mastication, and increased TMJ stress. Our study confirmed these trends, showing that the oldest age group exhibited significant reductions in multiple muscle thicknesses, further supporting the impact of sarcopenia on oral function.

### 4.3. Gender Differences in Muscle Morphology

Gender-based variations in muscle thickness have been well documented [14,17], with males generally exhibiting greater submental and masticatory muscle thickness compared to females. Ryu and Kim reported mylohyoid thickness ranging from 1.64–1.82 mm, with significantly higher values in males [14]. Miura et al. similarly found greater geniohyoid thickness in males, suggesting increased resistance to age-related atrophy and stronger jaw function [17].

Our findings align with these reports, as male participants demonstrated higher muscle thickness values across most measured muscles. These differences are clinically relevant because sex-based variations in muscle morphology may influence TMJ stability, occlusal function, and the risk of developing TMJ disorders over time. Future research should further explore gender-based adaptations in masticatory muscle function and their implications for oral rehabilitation strategies.

### 4.4. Neuromuscular Disorders and Muscle Morphology

Beyond normal aging and gender-related differences, neuromuscular diseases can significantly alter submental and masticatory muscle structure and function. Although patients with neuromuscular disorders were excluded from this study to minimize confounding variables, existing literature highlights their clinical importance [18]. Lagarde et al. reported that increased echogenicity of submental and masticatory muscles is observed in children with spastic cerebral palsy, indicating pathological muscle changes [19].

Given these findings, assessing masticatory and submental muscle morphology in neuromuscular disease populations is crucial, as their muscle characteristics may differ significantly from those of healthy individuals. Future studies should integrate muscle thickness assessments with electromyographic (EMG) data to better understand functional impairments in mastication and swallowing.

### 4.5. Comparing Imaging Modalities for Muscle and TMJ Evaluation

Various imaging modalities, including ultrasonography (US), MRI, and CT, are used to evaluate muscle morphology and TMJ function [17,19,20,21,22]. Each modality offers distinct advantages. MRI provides high-resolution soft tissue contrast, making it ideal for assessing muscle atrophy, inflammation, and TMJ disc displacement without radiation exposure [23,24]. CT is superior for visualizing bone structures, condylar morphology, and TMJ pathologies, but lacks soft tissue contrast. US is increasingly being used due to its real-time imaging capabilities, cost-effectiveness, and accessibility. It enables dynamic assessments of muscle function, structural asymmetry, and intervention guidance in oral and maxillofacial radiology [8,17,20,22]. While both MRI and US provide similar muscle thickness measurements, MRI remains the gold standard for evaluating deeper masticatory muscles and detecting subtle morphological changes [21,22]. Based on the preview studies, muscle thickness measurements varied in methodology, imaging modality, and sample size, influencing the comparability of findings. MRI-based studies, such as the study evaluating masseter and temporalis muscles in acromegaly patients [23] and the study assessing masticatory muscles in rheumatoid arthritis patients [24], employed high-resolution imaging for detailed soft tissue evaluation. These studies align with our methodology, as MRI allows for precise muscle thickness and quality assessments. Moreover, while US has been employed in previous muscle studies [8,16,17,19,25], MRI remains superior for deep muscle evaluation and structural integrity assessment, supporting its selection in our study. Future studies should integrate US with MRI to achieve a comprehensive analysis of both static muscle structure and dynamic function.

### 4.6. Muscle Morphology and TMJ Disorders

Our findings suggest that masticatory and submental muscle characteristics may influence TMJ structure and function, and it is essential to understand the functional and morphological muscle changes in individuals with TMD [5]. Previously, significantly higher activity values were observed for the sternocleidomastoid and trapezius muscles at rest in patients with TMD, and the magnitude of this activity was influenced by the presence of pain [26]. Previous research by Strini et al. demonstrated that masseter thickness varies between genders, with males exhibiting thicker muscles and greater bite force, consistent with our findings [5].

Additionally, Çebi et al. reported significant differences in masseter muscle thickness between affected and unaffected sides in patients with disc displacement with reduction, suggesting a role in occlusal imbalance and muscle asymmetry [27]. However, our study found no statistically significant difference between disc displacement and masseter asymmetry, indicating that other factors may contribute to TMJ structural changes.

### 4.7. Limitations

This study has several limitations that should be considered when interpreting the findings. First, the study lacks a control group, which limits direct comparisons between individuals with and without TMJ-related changes. Including a healthy control group in future studies would provide a clearer understanding of the impact of muscle morphology on TMJ structures. Second, the cross-sectional design of the study prevents the assessment of causal relationships between muscle thickness and TMJ conditions. Longitudinal studies would be necessary to evaluate progressive changes in muscle morphology and their clinical significance over time. Third, age-related muscle variations may have influenced the results, as sarcopenia and age-related muscle atrophy are known to affect submental and masticatory muscles. While our study categorized participants into different age groups, other confounding factors, such as genetic predisposition, occlusal habits, and overall muscle function, were not controlled for and may have contributed to the observed differences. Individual variations in muscle activity and TMJ loading were not assessed, which could influence muscle thickness measurements. Another limitation of this study is that tooth loss was not specifically evaluated as an exclusion criterion. Given that the absence of posterior teeth can alter masticatory biomechanics and TMJ loading, it is possible that variations in dental status may have influenced muscle thickness measurements. However, our study primarily focused on muscle morphology rather than occlusal function. Future studies should consider including occlusal status and specific patterns of tooth loss to further investigate their potential effects on masticatory muscle adaptation and TMJ morphology. Future research incorporating functional assessments, electromyographic studies, or additional imaging modalities may provide further insights into the relationship between muscle function, TMJ morphology, including occlusal status and specific patterns of tooth loss and associated disorders.

## 5. Conclusions

This study provides new insights into the relationship between submental and masticatory muscle morphology and TMJ structures, demonstrating significant associations between muscle thickness and disc–condyle position. These findings suggest that muscle characteristics may play a role in TMJ function and pathology, particularly in patients with disc displacement and condylar changes. While our results contribute to the clinical evaluation of patients with TMJ disorders, they should be interpreted as preliminary, given the cross-sectional design and absence of functional assessments. Future studies should build on these findings by incorporating dynamic imaging techniques such as functional MRI or ultrasound during jaw movement, as well as correlating imaging findings with clinical symptoms, bite force measurements, and electromyographic activity. Longitudinal studies are also needed to assess progressive muscle changes over time and their impact on TMJ disorders progression and treatment outcomes.

## Figures and Tables

**Figure 1 diagnostics-15-01535-f001:**
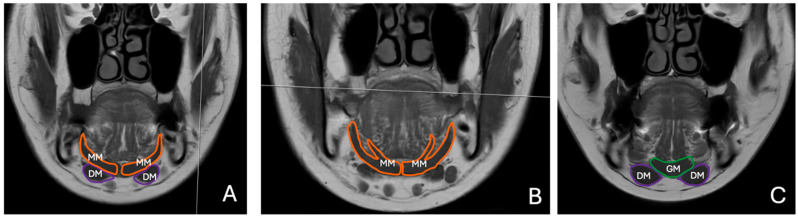
Coronal T1 FSE images showing submental muscles. (**A**) anterior belly of digastric (DM, purple) and mylohyoid (MM, orange) muscles, (**B**) anterior and posterior branches of mylohyoid muscle (MM, orange), (**C**) anterior belly of digastric (DM, purple) and geniohyoid (GM, green) muscles.

**Figure 2 diagnostics-15-01535-f002:**
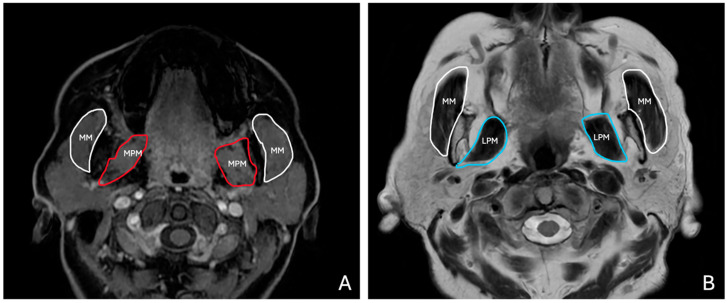
(**A**) Axial LAVA T1-weighted MR image shows masseter (MM, white) and medial pterygoid muscles (MPM, red). (**B**) T2-weighted FSE MR image shows masseter (MM, white) and lateral pterygoid muscles (LPM, blue).

**Figure 3 diagnostics-15-01535-f003:**
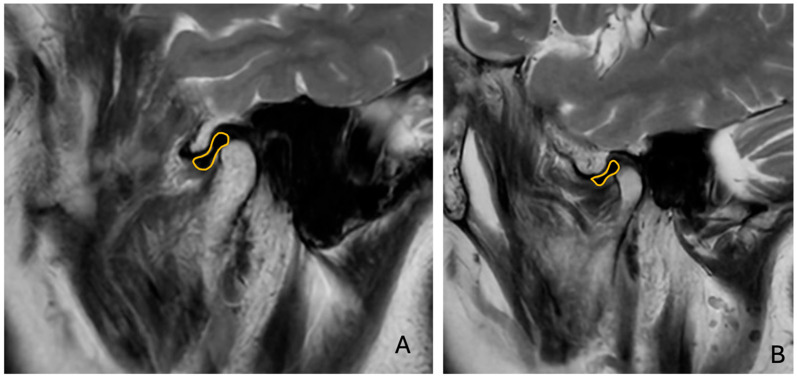
T2-weighted MR images of ((**A**) right and (**B**) left) disc displacement in closed-mouth position. Yellow areas show temporomanidbular disc.

**Figure 4 diagnostics-15-01535-f004:**
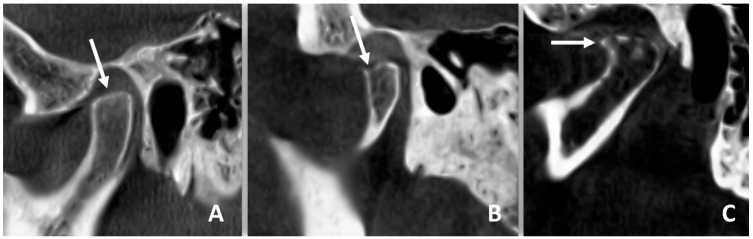
Computed tomography images of temporomandibular condyle evaluation showing flattening (**A**), erosion (**B**) and osteophyte formation (**C**). Arrows showing the cortex of mandibular condyle and the radiological finding.

**Figure 5 diagnostics-15-01535-f005:**
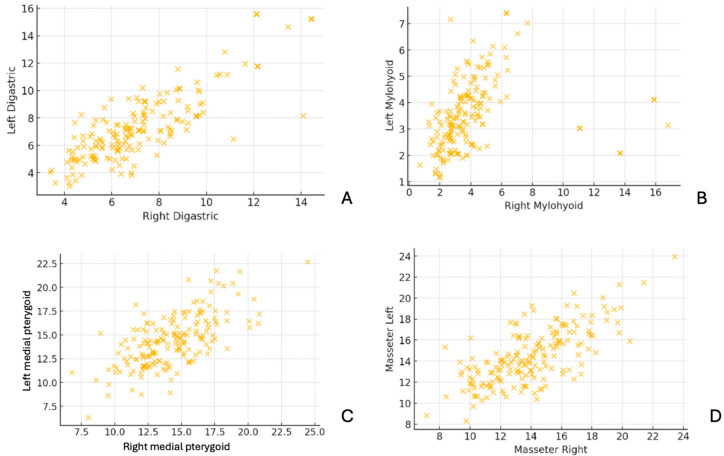
Scatter plots comparing right and left muscle thickness measurements. (**A**) Right vs. left digastric, (**B**) right vs. left mylohyoid, (**C**) right vs. left medial pterygoid, and (**D**) right vs. left masseter. Each point represents an individual measurement, showing the relationship between right and left muscle thickness across subjects. A strong positive correlation suggests symmetry in muscle thickness between the two sides.

**Figure 6 diagnostics-15-01535-f006:**
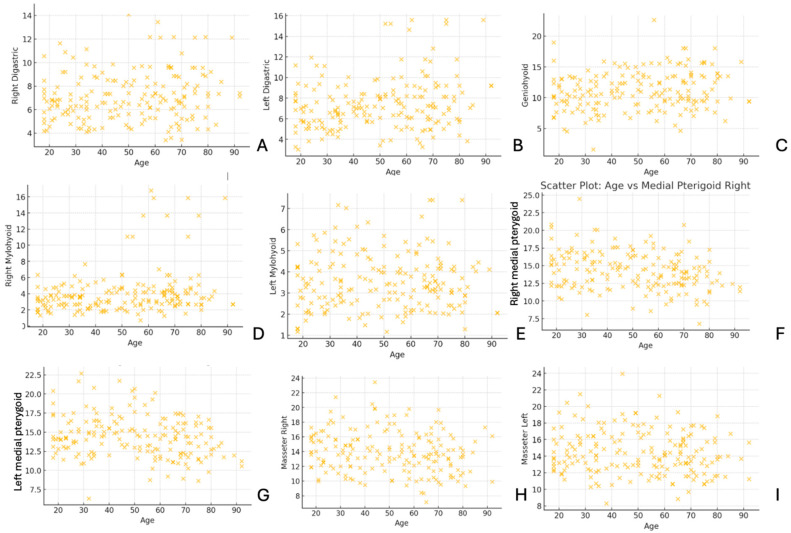
Scatter plots illustrating the relationship between age and muscle thickness measurements. (**A**) Right digastric, (**B**) left digastric, (**C**) geniohyoid, (**D**) right mylohyoid, (**E**) left mylohyoid, (**F**) right medial pterygoid, (**G**) left medial pterygoid, (**H**) masseter right, and (**I**) masseter left. Each point represents an individual measurement, showing variability in muscle thickness across different age groups.

**Table 1 diagnostics-15-01535-t001:** Mean thickness measurements (in mm) of submental and masticatory muscles in age groups.

Muscles	Age Groups			
	18–35	36–50	51–75	76–100
**Right digastric**	6.62	7.16	7.58	7.62
**Left digastric**	6.55	7.27	7.97	7.85
**Geniohyoid**	9.81	11.69	11.66	11.66
**Right mylohyoid**	3.31	3.59	4.61	4.93
**Left mylohyoid**	3.54	3.8	3.51	3.32
**Right medial pterygoid**	14.69	15.1	13.79	12.67
**Left medial pterygoid**	14.56	15.96	13.75	13.1
**Right masseter**	14.22	15.03	13.48	12.93
**Left masseter**	14.7	15.25	14.14	13.78
**Right lateral pterygoid**	14.69	15.1	13.81	12.64
**Left lateral pterygoid**	14.56	15.9	13.77	13.26

**Table 2 diagnostics-15-01535-t002:** Mean, minimum, and maximum thickness measurements (in mm) of submental and masticatory muscles in male and female participants.

Gender	Female			Male			Total		
	Mean	Min	Max	Mean	Min	Max	Mean	Min	Max
**Age**							50.086	18	92
**Right digastric**	7.03	3.45	14.43	7.48	3.4	14.43	7.21	3.4	14.43
**Left digastric**	7.11	3.02	15.58	7.82	3.28	15.58	7.4	3.02	15.58
**Geniohyoid**	10.81	1.6	22.62	11.67	4.64	18.99	11.16	1.6	22.62
**Right mylohyoid**	3.93	0.71	15.88	4.2	1.25	16.78	4.04	0.71	16.78
**Left mylohyoid**	3.47	1.17	7.4	3.63	1.33	7.4	3.53	1.17	7.4
**Right medial pterygoid**	13.67	6.78	20.84	15.17	8.6	24.46	14.28	6.78	24.46
**Left medial pterygoid**	13.85	6.31	21.68	15.14	8.64	22.69	14.38	6.31	22.69
**Right masseter**	13.56	7.14	20.48	14.52	8.35	23.43	13.95	7.14	23.43
**Left masseter**	14.16	8.83	20.44	14.86	8.3	23.94	14.44	8.3	23.94
**Right lateral pterygoid**	13.61	6.7	20.91	15.99	8.9	24.64	16.34	7.41	25.56
**Left lateral pterygoid**	13.92	7.67	22.44	16.54	8.78	25.62	15.41	6.34	22.91

**Table 3 diagnostics-15-01535-t003:** *p*-values for the correlation matrix, indicating the statistical significance of relationships between variables. Values below 0.05 denote significant correlations, while values above 0.05 suggest non-significant associations.

	Right Condyle	Right Disc–Condyle Position	Left Condyle	Left Disc–Condyle Position
**Right Digastric**	0.64	**0.02** *	0.205	0.369
**Left Digastric**	0.405	0.09	0.469	0.915
**Geniohyoid**	0.568	0.95	0.471	0.402
**Right Mylohyoid**	0.254	**0.004** *	0.09	0.875
**Left Mylohyoid**	0.489	0.799	0.136	0.080
**Right Medial Pterygoid**	0.476	0.781	0.132	0.765
**Left Medial Pterygoid**	0.339	0.849	**0.02** *	0.571
**Right Masseter**	0.356	0.06	0.135	0.839
**Left Massseter**	0.06	0.112	**0.014** *	0.962
**Right Lateral Pterygoid**	0.496	0.711	0.132	0.761
**Left Lateral Pterygoid**	0.378	0.889	**0.02** *	0.536

Values marked with * indicate statistically significant correlations (*p* < 0.05).

## Data Availability

Data have been reported in the manuscript and can be found in the manuscript.

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
