# Peer review of "Magnetic Resonance Imaging of Submental and Masticatory Muscle Morphology and Its Relationship with Temporomandibular Joint Structures"

_diagnostics, 2025, doi:10.3390/diagnostics15121535_

Round 1

Reviewer 1 Report

Comments and Suggestions for Authors

The study is a valuable contribution and is generally well-executed. The following actionable recommendations are offered to improve the manuscript:

  1. The authors should justify why they grouped patients by age and explain whether they expect age-related sarcopenia to affect muscle thickness.
  2. The authors should explicitly state the reference criteria for diagnosing disc displacement on MRI.
  3. Address additional limitations, like the lack of a control group and the cross-sectional nature as limiting factors. Also acknowledge the potential influence of age or other confounders on the results. This can be done in a dedicated limitations paragraph for clarity.
  4. The study provides only a brief mention of the advantages and limitations of other imaging modalities. This discussion should be expanded, particularly regarding ultrasound, which is highly accessible, cost-effective, and increasingly utilized in clinical practice for evaluating masticatory and submental muscles. Given its ability to assess real-time muscle function, asymmetry, and structural changes, ultrasound serves as a valuable complement to MRI, especially in functional and interventional applications. Please see this paper: Popescu MN, Beiu C, Iliescu CA, Racoviță A, Berteanu M, Iliescu MG, Stănescu AMA, Radaschin DS, Popa LG. Ultrasound-Guided Botulinum Toxin-A Injections into the Masseter Muscle for Both Medical and Aesthetic Purposes. Toxins. 2024; 16(10):413. https://doi.org/10.3390/toxins16100413
  5. In the conclusion section, the authors might slightly broaden the applicability by saying these findings could help in clinical evaluation of TMD patients, but also emphasize that they are preliminary. The suggestion for future studies could be even more specific. For example, “future studies should include dynamic MRI or ultrasound during function, and correlate imaging findings with clinical symptoms and bite force measurements.” 

Author Response

Dear Reviewer, Thank you for your valuable comments. The suggested revisions are completed and you can kindly find them in the attached file. 

Reviewer 2 Report

Comments and Suggestions for Authors

Dear authors

The objective of the study is partially answered. This includes using CT images to assess bone tissue, which is not demonstrated in the methodology and is not quantified in the results.

Your sample of 185 patients ' ages ranged from 18 to 95 years. Although it is stratified by age group, we can´t observe this in the results. In table 1, it is done by gender but does not take age into account, which may be relevant because the biomechanics of the temporomandibular joint changes.

As for the exclusion criteria, the lack of teeth remains to be established, and which ones, considering that their absence will modify the biomechanics of the temporomandibular joint.

There is no evidence regarding the results obtained from the analysis of CT images; The images used in the text are MRI (figure 1-3).

Regardless of who performed the measurements, the inter and intra observer measurements should be validated in a reduced sample, for example, n=20.

Figures 4 and 5 are not introduced in the body of the text.

The conclusions go beyond the data obtained.

The references used are adequate.

In conclusion, the analysis should consider age groups and the use of CT.

Author Response

(The authors gave the same response as above.)

Round 2

Reviewer 2 Report

Comments and Suggestions for Authors

Dear authors, the changes have addressed some of the questions/considerations raised in the first review. Tables should be standardized in their presentation, such as in font size and type (format). The text concerning each figure should precede it. 

The discussion could eventually be improved; it is sometimes confusing.

Congratulations on the work done!

Author Response

Dear reviewer, thank you for your valuable and supportive comments, you can find detailed explanations in the PDF file attached. 

Thank you 
